# MicroRNAs Possibly Involved in the Development of Bone Metastasis in Clear-Cell Renal Cell Carcinoma

**DOI:** 10.3390/cancers13071554

**Published:** 2021-03-28

**Authors:** Lisa Kinget, Eduard Roussel, Diether Lambrechts, Bram Boeckx, Loïc Vanginderhuysen, Maarten Albersen, Cristina Rodríguez-Antona, Osvaldo Graña-Castro, Lucía Inglada-Pérez, Annelies Verbiest, Jessica Zucman-Rossi, Gabrielle Couchy, Stefano Caruso, Annouschka Laenen, Marcella Baldewijns, Benoit Beuselinck

**Affiliations:** 1Department of General Medical Oncology, Leuven Cancer Institute, University Hospitals Leuven, 3000 Leuven, Belgium; lisa.kinget@uzleuven.be (L.K.); loic.vanginderhuysen@uzleuven.be (L.V.); Annelies.Verbiest@uzleuven.be (A.V.); 2Department of Urology, University Hospitals Leuven, 3000 Leuven, Belgium; eduard.roussel@uzleuven.be (E.R.); maarten.albersen@uzleuven.be (M.A.); 3Laboratory of Translational Genetics, Department of Human Genetics, KU Leuven, 3000 Leuven, Belgium; Diether.Lambrechts@vib-kuleuven.be (D.L.); bram.boeckx@kuleuven.be (B.B.); 4VIB Center for Cancer Biology, VIB, 3000 Leuven, Belgium; 5Centro Nacional de Investigaciones Oncológicas (CNIO), 28040 Madrid, Spain; crodriguez@cnio.es (C.R.-A.); ograna@cnio.es (O.G.-C.); 6Department of Statistics and Operational Research, Faculty of Medicine, Complutense University, 28040 Madrid, Spain; lucia.inglada.perez@ucm.es; 7Centre de Recherche des Cordeliers, Sorbonne Université, Université de Paris, INSERM, Functional Genomics of Solid Tumors Laboratory, Équipe Labellisée Ligue Nationale contre le Cancer, Labex OncoImmunology, F-75006 Paris, France; jessica.zucman-rossi@inserm.fr (J.Z.-R.); gabrielle.couchy@inserm.fr (G.C.); stefano.caruso@inserm.fr (S.C.); 8Department of Biostatistics, KULeuven, 3000 Leuven, Belgium; annouschka.laenen@kuleuven.be; 9Department of Pathology, University Hospitals Leuven, 3000 Leuven, Belgium; marcella.baldewijns@uzleuven.be

**Keywords:** renal cell carcinoma, bone metastasis, microRNA, miR-23a-3p, miR-20a-5p, miR-27a-3p, miR-335-3p, miR-30b-3p, miR-139-3p, SATB2

## Abstract

**Simple Summary:**

Bone metastases cause substantial morbidity and implicate worse clinical outcomes for clear-cell renal cell carcinoma patients. MicroRNAs are small RNA molecules that modulate gene translation and are involved in the development of cancer and metastasis. We identified six microRNAs that are potentially specifically involved in metastasis to bone, of which two seem protective and four implicate a higher risk. This aids further understanding of the process of metastasizing to bone. Furthermore, these microRNA hold potential for biomarkers or therapeutic targets.

**Abstract:**

Bone metastasis in clear-cell renal cell carcinoma (ccRCC) leads to substantial morbidity through skeletal related adverse events and implicates worse clinical outcomes. MicroRNAs (miRNA) are small non-protein coding RNA molecules with important regulatory functions in cancer development and metastasis. In this retrospective analysis we present dysregulated miRNA in ccRCC, which are associated with bone metastasis. In particular, miR-23a-3p, miR-27a-3p, miR-20a-5p, and miR-335-3p specifically correlated with the earlier appearance of bone metastasis, compared to metastasis in other organs. In contrast, miR-30b-3p and miR-139-3p were correlated with less occurrence of bone metastasis. These miRNAs are potential biomarkers and attractive targets for miRNA inhibitors or mimics, which could lead to novel therapeutic possibilities for bone targeted treatment in metastatic ccRCC.

## 1. Introduction

Bone metastases (BM) occur in about 22% to 35% of all metastatic clear-cell renal cell carcinoma (ccRCC) patients [1,2]. They can give rise to skeletal related events (SREs) such as pain, hypercalcemia, pathologic fractures, and myelum compression, causing substantial morbidity. Moreover, BM are associated with poor prognosis in ccRCC when compared to metastases in other organs, and their presence also predicts an inferior outcome in treatment with angiogenesis inhibitors [3,4,5].

Preventing the occurrence of BM in ccRCC patients would provide substantial clinical benefit. Bone resorption inhibitors such as bisphosphonates or denosumab can reduce the number of SRE, though their impact on outcome is not precisely defined [6,7,8]. In order to provide additional therapeutic options to limit the occurrence of BM in ccRCC, better understanding of the underlying molecular mechanisms is required. Previously, we demonstrated that higher receptor activator of nuclear factor κB (RANK) and lower osteoprotegerin (OPG) expression in the primary kidney tumor are correlated with the occurrence of BM [9], findings that were similar to those of Mikami et al. [10].

MicroRNAs (miRNAs) are a class of small non-protein-coding RNA molecules that are principally involved in posttranscriptional regulation by interfering with translation or stability of target transcripts. Through imperfect base pairing, a single miRNA can regulate a large variety of gene transcripts. Dysregulation of miRNAs is frequently associated with cell proliferation, metastasis and apoptosis in various cancers. Several miRNA are currently under development as targets for cancer therapies [11,12,13,14].

In breast cancer (BC) and prostate cancer (PC), diseases with mainly osteoblastic BM, the involvement of miRNAs in the development of BM is well documented. There is less evidence in ccRCC, a malignancy with mainly osteolytic BM. Therefore, our aim was to study the impact of miRNAs on the development of BM in ccRCC.

## 2. Materials and Methods

### 2.1. Patient Selection

We used clinical and molecular data available in the University Hospitals Leuven database of ccRCC patients treated with systemic therapies. All patients underwent nephrectomy as first therapeutic intervention.

The study was approved by the Ethics Committee Research UZ/KULeuven (approval number S53479/S63833). Signed consent was obtained from all patients. In some cases, we used biological material from patients who were deceased and for whom general positive advice for the utilization of remaining tissue was foreseen by the institutional board.

### 2.2. Study Objectives and Endpoints

The principal objective was to investigate miRNA involvement in the development of BM in ccRCCs. Therefore, the primary endpoint of the study was time-to-bone-metastasis (TTBM), defined as the time between initial diagnosis and development of BM. In order to evaluate if the impact on metastasis was specific for BM, we calculated time-to-metastasis-other-than-bone (TTMOTB), defined as the time between initial diagnosis and the development of metastasis in all organs other than BM (MOTB). Furthermore, we included an explorative analysis of the impact of miRNAs on mRNA levels. As BM have a negative prognostic impact in ccRCC, our third objective was to study the impact of the miRNAs involved in the development of BM on clinical outcome, including overall survival (OS), since diagnosis and outcome on first-line anti-vascular endothelial growth factor receptor tyrosine kinase inhibitor (VEGFR-TKI) as measured by response rate (RR), progression free survival (PFS) and OS since start of VEGFR-TKIs.

### 2.3. MiRNA Extraction

MiRNA analysis was performed at the Centro Nacional de Investigaciones Oncológicas in Madrid as described previously [15]. For miRNA isolation, H&E-stained sections of the tumor samples were examined by a pathologist to confirm the diagnosis and to estimate tumor content. Total RNA was isolated with the Recover All Total Nucleic Acid Isolation kit (Ambion) using 4–6 whole 10-μm sections of the tumor samples. RNA quantity and quality were assessed by NanoDrop Spectrophotometer (Nanodrop Technologies, Wilmington, DE, USA). Quantitative RT-PCR analysis was performed using the Universal miRCURY LNATM microRNA PCR System (Exiqon, Vedbaek, Denmark), following the manufacturer’s instructions. In brief, 10 ng of total RNA was reversely transcribed, using the miRCURY LNATM Universal RT microRNA PCR kit (Exiqon). The resulting cDNA was diluted and PCR reactions were carried out using ExiLENT SYBR^®^ Green Master Mix kit (Exiqon) and LNA™-enhanced microRNA PCR primer sets (Exiqon). MiRNA expression was quantified using the QuantStudio™ 6–7 Flex Real-Time PCR System (Applied Biosystems, Foster City, CA, USA). Stably expressed endogenous control 5s RNA was used for data normalization. Three samples with a low expression of the control 5s RNA were excluded from the analysis. Expression of each miRNA was calculated using the comparative Ct method using the Ct value of the endogenous control to normalize the data. Negative controls were present in all PCR series and assays were carried out in quadruplicates. The expression level of 454 miRNAs was defined in each tumor sample.

### 2.4. mRNA Extraction

Archived formal-fixed paraffin-embedded (FFPE) samples were retrieved. Tissue blocks with the highest tumoral content were selected for further processing through review of H&E slides. Blank slides with 5 μm thickness were made and underwent macrodissection to exclude non-tumoral tissues, resulting in a total tumoral surface area of 50–1800 mm^2^. The Maxwell RSC RNA FFPE kit (Promega) was used to perform RNA extraction according to the manufacturer’s instructions. The Forward QuantSeq 3′ mRNA-Seq Library Prep Kit for Illumina (Lexogen) was used to prepare cDNA libraries according to the manufacturer’s instructions with 5 µL of RNA as input and 16 PCR cycles. cDNA concentrations and fragment length were measured with the QubitTM dsDNA HS assay (Thermofisher, Waltham, MA, USA) and Bioanalyzer HS DNA electrophoresis (Agilent, Santa Clara, USA). Clonal clusters were generated with Illumina cBOT. The HiSeq 4000 kit (Illumina) was used for RNAseq according to the manufacturer’s instructions. Raw sequencing reads were trimmed of adaptors and optical duplicates and were subsequently aligned to the human reference genome hg19 with HiSat2 (v2.1.0) and quantified using featureCounts (v1.6.4). Counts were processed using DESeq2 (v1.26.0) and normalized using the VarianceStabilizingTransformation function.

### 2.5. Selection of miRNAs and Genes

We selected 78 miRNA with previously described involvement in either bone metabolism, oncogenesis, or metastasis (Table 1). For the estimation of TTBM, we selected the subset of patients (*n* = 106) who developed BM after the initial diagnosis of the ccRCC (metachronous BM), as TTBM was uncertain in patients with BM at diagnosis. For the estimation of TTMOTB, we selected the subset of patients who developed metastasis in organs other than bone after the initial diagnosis (*n* = 72). Patients with BM at initial diagnosis, but no metastases at other sites, were also included for TTMOTB estimations. We selected 29 genes with known involvement in either bone metabolism, oncogenesis or metastasis (Table 1) for a correlation analysis between miRNA and mRNA expression, for which Spearman correlation was used. For this analysis, we used mRNA and miRNA expression data from all the tumor samples, independently of the timing of development of metastases. As we performed this study from a validation perspective, results with *p*-value < 0.05 were considered significant and no correction for multiple testing was performed. Correlated miRNA-gene pairs were subsequently searched in DIANA TarBase v8, a database of experimentally validated miRNA-target pairs [16], and miRDB, a database for prediction of functional miRNA targets [17].

### 2.6. Clinical Data

After nephrectomy, patients underwent follow-up medical imagery (with either abdominal ultrasonography and chest X-ray or computed tomography of the chest and abdomen) every three to six months. In patients treated with VEGFR-TKIs, follow-up computed tomography (chest and abdomen) was done every two to three months during treatment. The decision whether or not to perform bone scintigraphy, a modality with low sensitivity to detect osteolytic BM, was made on clinical grounds. Pathology slides were reviewed by expert genitourinary pathologists. The International Metastatic RCC Database Consortium (IMDC) prognostic score was retrospectively calculated for each patient [84].

### 2.7. Statistical Analysis

Descriptives for time-to-event variables are based on Kaplan–Meier estimates. Cox proportional hazards regression models are used for the analysis of the association between expression data and time-to-event outcomes. Analyses were performed using SAS software (version 9.4) (SAS Institute Inc., Cary, NC, USA) and R (version 4.0.03) (R Core Team, Vienna, Austria).

## 3. Results

### 3.1. Included Patients

In total, the data of 128 patients were available. Patient characteristics are described in Table 2. All patients reached the metastatic stage and were treated with systemic therapies, among them 109 with VEGFR-TKIs in first-line. Figure 1 shows the occurrence of BM and MOTB over time in the patient cohort. Patients with synchronous BM or MOTB were excluded for analysis of TTBM or TTMOTB, respectively, resulting in patient groups of 106 and 72, respectively. For the OS analysis, all patients were included.

### 3.2. MiRNAs Associated with Longer or Shorter Time-to-Bone-Metastasis 

A total of 4 out of 78 miRNA were significantly associated with longer TTBM: miR-204-5p, miR-30b-3p, miR-542-5p, and miR-139-3p. miR-204-5p and miR-542-5p were also correlated to TTMOTB, thus their effect might not be specific for BM. miR-30b-3p and miR-139-3p were not correlated with TTMOTB, thus their effect might be specific for BM.

Nine miRNAs were found to be significantly associated with shorter TTBM: miR-21-5p, miR-21-3p, miR-28-3p, miR-34c-5p, miR-23a-3p, miR-20a-5p, miR-335-3p, miR-182-5p, and miR-27a-3p. However, miR-21-5p, miR-34c-5p, miR-28-3p, and miR-21-3p were also correlated to TTMOTB, thus their effect might not be specific for BM. miR-182-5p showed a trend towards correlation with TTMOTB. miR-20a-5p, miR-335-3p, miR-27a-3p, and miR-23a-3p were clearly not correlated to TTMOTB, thus their effect might be specific for the development of BM.

The 13 miRNAs are displayed with hazard ratio (HR) and confidence interval (CI) in Figure 2. The correlation results for all miRNA are presented in Appendix A.

### 3.3. Correlation of miRNA with Gene Expression 

Using the Spearman correlation, we correlated the expression of 13 miRNAs that are correlated to TTBM, with the expression of 29 mRNAs. mRNA expression data were available for 89 of the 128 patients. A non-systematic literature search was able to validate 13 previously described correlations between miRNAs and mRNAs.

Among the miRNAs associated with longer TTBM, miR-204-5p was inversely correlated with RUNX2 and TGFB1. miR-139-3p was correlated with the mRNA expression levels of NOTCH1. miR-30b-3p was inversely correlated with the mRNA expression levels of CXCL8. miR-542-5p was correlated with the mRNA expression levels of SMAD2.

Among the miRNAs associated with shorter TTBM, miR-21-3p was associated with reduced FOS, BMPR2, and SMAD4 expression and increased TFGB1 expression. miR-34c-5p was correlated with reduced SATB2 levels. miR-20-5p was associated with upregulated RUNX2 levels. Finally, miR-182-5p was correlated with lower SATB2 levels.

We studied other possible correlations between miRNAs and mRNAs in an exploratory approach. The significant findings are represented in Table 3. All correlation results are available in Appendix A.

### 3.4. Correlation of miRNA with Clinical Outcome 

Subsequently, we checked the impact of the 13 miRNA correlated with TTBM, on outcome. Several of these miRNAs were also correlated with OS (Table 4). miR-204-5p, associated with longer TTBM, is significantly correlated with longer OS. miR-21-5p, miR-21-3p, miR-34c-5p, miR-335-3p, and miR-182-5p, associated with shorter TTBM, were significantly correlated with shorter OS. The full correlation results are available in Appendix A.

As the majority of the patients (*n* = 109) were treated in first-line with VEGFR-TKIs, we compared the expression levels of these 13 miRNA with PFS and OS on VEGFR-TKIs (Appendix A). miR-204-5p, correlated with longer TTBM, was also correlated with longer PFS and OS on VEGFR-TKIs. miR-21-3p and miR-34c-5p, correlated with shorter TTBM, were also correlated with shorter PFS and OS on VEGFR-TKIs. Finally, miR-21-5p and miR-182-5p, correlated with shorter TTBM, were correlated with shorter OS. miR-204-5p was associated with increased RR (*p* = 0.03) and miR-34c-5p with decreased RR (*p* = 0.003).

Additionally, SATB2 was associated with longer PFS (HR = 0.595; *p* = 0.045) and longer OS on VEGFR-TKI (HR = 0.595; *p* = 0.039).

### 3.5. Correlation of mRNA Expression with TTBM and OS 

For only 71 of the 106 patients eligible for TTBM analysis, mRNA expression data were available. On this reduced patient series, two significant correlations with TTBM were found: lower SATB2 (*p* = 0.03) and higher IL11 (*p* = 0.048) mRNA expression were correlated with shorter TTBM.

There were 89 patients available for the analysis of the correlation between mRNA expression and OS. Higher SATB2 (*p* < 0.0001), FOS (*p* = 0.02) and SMAD4 (*p* = 0.02) mRNA expression were associated with longer OS. Higher CD44 (*p* < 0.001), ITGA5 (*p* = 0.004), TGFB1 (*p* = 0.008), TGIF2 (*p* = 0.01), SRCIN1 (*p* = 0.02), RUNX2 (*p* = 0.02), CHD11 (*p* = 0.04), and ITGA3 (*p* = 0.04) were correlated with worse OS. Full results are presented in Appendix A.

## 4. Discussion

Bone metastases in metastatic ccRCC patients occur in up to 35% of patients [1,2]. Several studies have reported a negative impact of BM on clinical outcomes [3,95]. Within ccRCC patients with BM, those with synchronous BM are generally poor risk patients, whereas patients with metachronous BM and longer TTBM were associated with longer OS [96]. To provide new therapeutic options for BM in ccRCC, further research into their pathogenesis is critical. miRNA have attracted much attention in the past years and therapeutic strategies that target them are already used in phase I clinical trials [11,12,13].

Most evidence about the involvement of miRNA in BM comes from BC and PC, malignancies with mostly osteoblastic metastasis. Of osteolytic metastasis, most evidence is in lung and colorectal cancer (CRC). In ccRCC, the amount of evidence is more limited.

Therefore, we studied the impact of miRNAs expressed in primary ccRCC tumors on the development of BM. We report 13 miRNAs that are significantly associated with TTBM. Of these, six miRNA did not have a significant correlation with TTMOTB and might therefore be specific for osteotropic metastasis. We also report correlations between the intratumoral expression of miRNAs and mRNA of genes involved in invasion and metastasis and more specifically in the development of BM. As miRNAs influence gene expression through translation repression or increased degradation of target mRNA, inverse correlations could result from either direct mRNA targeting or from a target upstream in its regulatory pathway. We also reported positive correlations as these could indirectly result from targeting suppressive components of regulatory pathways of the correlated gene. Finally, several miRNAs that are correlated to the development of BM were also associated with outcome.

### 4.1. miRNA Associated with Longer TTBM

In this study, we identified four miRNA associated with longer TTBM (miR-30b-3p, miR-139-3p, miR-204-5p and miR-542-5p). miR-30b-3p and miR-139-3p were not significantly associated with TTMOTB and might thus be specific for the development of BM. All four miRNAs have demonstrated tumor suppressive properties over a variety of cancer types.

In our series, **miR-30b-3p** was associated with longer TTBM and inversely correlated to mRNA expression of several genes involved in BM (CD44, ITGA3, TCF7, RANKL and CXLC8). The miR-30 family, to which miR-30b-3p belongs, suppresses BM in BC and PC. These miRNAs inhibit osteoblast differentiation and osteomimicry by targeting RUNX2 [54,97,98,99]. In BC cells, miR-30 inhibition enhances tumorigenesis and metastasis by targeting UBC9 and ITGB3 and osteomimicry of cancer cells by targeting RUNX2, ITGA5 and CDH11 [100]. In BC, miR-30b-3p suppresses BM by targeting CXCL8, IL11, DKK1, RUNX2, CDH11, CTGF, ITGA5, and ITGB3 [36]. Finally, in multiple myeloma, an osteolytic disease, miR-30b-3p has a tumor suppressor role targeting the Wnt/b-Catenin/BCL9 Pathway [101].

We have found that **miR-542-5p** was associated with longer TTBM and TTMOTB, inversely correlated with the mRNA expression of genes involved in BM (CD44, CXCR4, RUNX2 and TCF7) and positively correlated with the mRNA expression of genes protective against BM (SATB2 and SMAD2). miR-542-5p acts as a tumor suppressor in non-small cell lung cancer (NSCLC) [67], neuroblastoma [102], osteosarcoma [103], CRC [104], BC [105], and hepatocellular carcinoma (HCC) [106], though contradictory evidence exists [107]. A role in bone metabolism has been described for the 3p strand of the miR-542 duplex, as a suppressor of osteoblast cell proliferation and differentiation by targeting BMP7-signalling [66]. miR-542-5p has been shown to increase expression levels of SMAD2 in other diseases [88].

In our series, **miR-139-3p** was associated with longer TTBM, inversely correlated with the mRNA expression of genes involved in BM (TCF7 and ITGA3), and positively correlated with the mRNA expression of genes protective for BM (BMPR2 and NOTCH1). Additionally, it is also correlated with longer OS. miR-139-3p and its guide strand miR-139-5p are involved in ccRCC pathogenesis and expression levels of both miRNA and several of their target genes are correlated with clinical outcomes [108]. A tumor suppressor role has also been described in bladder cancer [109], CRC [110], glioblastoma multiforme [111], BC [45], and HCC [112]. Additionally, miR-139-3p is downregulated in serum in patients with esophagal squamous cell cancer (SCC), highlighting its potential as a serum biomarker [113]. miR-139-5p is a regulator of osteoblast differentiation and apoptosis by targeting ELK1 and ODSM [44]. In NSCLC, miR-139-5p serum levels are reduced in the presence of lytic BM when compared to metastasis at other sites. Increased expression of miR-139-5p enhances osteogenic differentiation of mesenchymal stem cells (MSC) through NOTCH1 signaling. After exposure to NSCLC cells, this pathway was downregulated [46]. Conditioned media from NSCLC cells have been shown to promote osteoclastogenesis through inhibition of miR-139 and activation of the STAT3/c-FOS/NFATc1 pathway [114]. Activation of the Notch signaling pathway is involved in oncogenesis in a variety of malignancies including ccRCC [115].

Finally, **miR-204-5p** was associated with longer TTBM and TTMOTB, inversely correlated with the expression of genes involved in BM (CD44, RANKL, ITGA3, RUNX2, CDH11, TCF7, ITGA5, TGFB1, CXCL8 and CTGF), correlated with the expression of genes protective for BM (FOS, BMPR2, SATB2 and SMAD4) and correlated with longer OS. miR-204-5p is usually downregulated in ccRCC tumor tissues [116,117,118] and seems protective for the development of BM. Moreover, miR-204-5p is lower in the more aggressive ccrcc1+4 molecular ccRCC subtypes compared to the less aggressive ccrcc2+3 subtypes [119]. In bone metastatic BC cells, miR-204-5p inhibits TGF-β-induced IL11 production [59]. In laryngeal SCC, miR-204-5p inhibits cell proliferation, invasion, and metastasis [120]. In BC cells, miR-204-5p inhibits viability, proliferation and migration [121]. In oral SCC, it inhibits cell proliferation and metastasis by targeting CXCR4 [122]. Finally, miR-204-5p was shown to inhibit osteogenic differentiation of MSC [123].

### 4.2. miRNA Associated with Shorter TTBM

In our series, nine miRNA were associated with shorter TTBM. Of these, most were also associated with shorter TTMOTB, however miR-23a-3p, miR-20a-5p, miR-335-3p, and miR-27a-3p were not and might therefore be specifically involved in the development of BM.

We show that **miR-23a-3p** expression was correlated with TTBM. The oncogenic properties of miR-23a-3p have been previously demonstrated. In ccRCC tissues and cell lines, where it is overexpressed, it blocks apoptosis and enhances proliferation and mobility. Its upregulation in ccRCC is associated with worse OS [33]. Moreover, several studies in gastro-intestinal malignancies [124,125] and BC [126] have highlighted its potential as a serum biomarker. It is part of the miRNA cluster 23a/27a/24-2, which is involved in oncogenesis and inhibits osteoblast differentiation by targeting RUNX2 and SATB2 [34]. Forced expression of this miRNA cluster enhances cell migration and metastasis in BC cells [127]. Inhibition of miR-23a-3p promotes osteoblast proliferation and differentiation [32]. miR-23a-3p also inhibits osteogenic differentiation of human MSC [128]. Surprisingly, in our tumor series, miR-23a-3p expression was inversely correlated with ELK1 and CXCL8 expression and positively correlated with BMPR2 expression. This seems contradictory, as BMPR2 expression is inversely correlated with metastasis occurrence in prostate BM [70].

In our tumor series, **miR-20a-5p** expression was correlated to TTBM, inversely correlated to the expression of the BM protective gene OPG and correlated to the expression of several genes involved in BM (RUNX2, TCF7 and IL11). In ccRCC, miR-20 has a higher expression in BM compared to primary tumor and to normal renal tissues [27]. Additionally, increased serum expression levels in gastric cancer and CRC show its potential as a biomarker [129,130]. The positive correlation between RUNX2 and miR-20a-5p was previously described in adipose stem cells during osteogenic differentiation. MiR-20a promotes the osteogenesis of MSC through targeting of PPARγ, BAMBI, and CRIM1, which are negative regulators of BMP signaling [93]. Exosomal miR-20a-5p derived from BC cells promotes proliferation and differentiation of osteoclasts [28]. The inverse correlation with OPG, a protein which acts as a decoy receptor for RANK and neutralizes its role in osteoclastogenesis [131], seems biologically consistent with the association of miR-20a-5p with shorter TTBM. Previous reports have however described a positive correlation with miR-20a during osteogenic differentiation [93]. Surprisingly, ITGB3 expression was inversely and PDCD4 expression positively correlated with miR-20a-5p. In bladder and CRC cells, PDCD4 is downregulated by miR-20a [132,133].

We observed that **miR-27a-3p** was associated with TTBM and inversely correlated with expression of the bone protective gene BMP7. In ccRCC, miR-27a-3p promotes proliferation and metastasis [134,135]. In osteosarcoma, CRC, pancreatic, and gastric cancer, a pro-oncogenic role is also seen [136,137,138,139,140], although in NSCLC [141] and HCC [142] a tumor suppressive function has been proposed. Serum levels of miR-27a-3p are elevated in pancreatic cancer, CRC, and PC, demonstrating its potential as a non-invasive biomarker [143,144,145]. miR-27a-3p is involved in bone metabolism and negatively regulates osteogenic differentiation [146]. It is part of the aforementioned 23a/27a/24-2 miRNA cluster, which is involved in both oncogenesis and osteoblast differentiation by targeting RUNX2 and SATB2 [34]. Surprisingly, miR-27a-3p was also inversely correlated with CXCL8, promotor of osteoclastogenesis, [76] and positively correlated with PDCD4 expression.

In our tumor series, **miR-21-3p and miR-21-5p** were correlated with TTBM and TTMOTB. miR-21-3p was inversely correlated with the expression of several BM protective genes (FOS, BMPR2, SMAD4 and SATB2) and correlated with the expression of several genes involved in BM (CXCL8, TGFB1, CD44, RANKL and ITGA3). miR-21-5p expression was inversely correlated with the expression of two genes protective for BM (SMAD4 and SATB2) and correlated with the expression of genes involved in BM (TCF7, CD44, and ITGA3). miR-21-5p and miR-21-3p were significantly correlated with shorter OS. Both the 3p and 5p strand of miR-21 are known pro-oncogenic miRNAs across tumor types such as NSCLC, gastric cancer, CRC, and BC [147,148,149]. In ccRCC tissues, miR-21 is upregulated [118,150] and higher expression is also correlated with worse clinical outcomes [151]. In our previous study, miR-21-3p and miR-21-5p were upregulated in the more aggressive ccRCC molecular ccrcc1+4 subtypes [119]. miR-21 plays a pivotal role in RANKL-induced osteoclastogenesis through regulation of PDCD4 and OPG, which is critical for the development of lytic BM [30,71,152]. Exosomal miR-21 from NSCLC cells facilitates osteoclastogenesis by targeting PDCD4 [30] and inhibition of miR-21 impairs osteoclast activity in multiple myeloma by targeting OPG and PIAS3 [152].

In our samples, **miR-34c-5p** was associated with shorter TTBM and TTMOTB, correlated with the expression of genes involved in BM (RANK, TGFB1, CD44 and ITGA3), inversely correlated with the expression of genes protective for BM (FOS and SATB2) and significantly correlated with shorter OS. miR-34 inhibits osteoblast proliferation and differentiation in the mouse by targeting SATB2 [92]. In bladder cancer, miR-34c-5p enhances proliferation and migration of bladder cancer cells through targeting of NOTCH1 [153]. However, in other malignancies such as NSCLC and laryngeal SCC, a tumor suppressive role is observed [154,155] and a phase I clinical trial has been conducted with a miR-34a mimic in advanced solid tumors [11]. In our previous study, miR-34c-5p was correlated with the more aggressive ccrcc1+4 molecular subtypes [119].

We found that **miR-335-3p** was correlated with shorter TTBM, inversely associated with OPG expression, positively associated with the expression of genes involved in BM (RANKL, ITGA3, ITGA5, TGFB1, RUNX2, CD44, TCF7 and CDH11), and associated with shorter OS. Unlike this impressive association with the expression of several genes involved in metastasis and BM, the role of miR-335-3p remains unclear. In our previous study, miR-335-3p was correlated with the more aggressive ccrcc1+4 molecular subtypes, [119] but evidence on its prognostic effects remains conflicting [156,157,158]. A pro-oncogenic role was described in astrocytoma [159], but a tumor suppressor role described in BC [160]. In small cell lung carcinoma xenograft models and in PC, miR-335-3p overexpression was protective for BM, however, BM in these diseases are osteoblastic [68,161].

In our tumor series, **miR-182-5p** expression was correlated with shorter TTBM and TTMOTB, inversely correlated with the expression of genes protective for BM (SATB2, OPG and BMPR2), correlated with the expression of several genes involved in BM (IL11, CDH11, RUNX2, CXCL8, TCF7, CD44, and ITGA3), and correlated with shorter OS. In the literature, conflicting evidence exists for miR-182-5p, as previous studies suggest a tumor suppressive role for this miRNA in ccRCC [162,163,164]. In our previous study, miR-182-5p was correlated with the more aggressive ccrcc1+4 molecular subtypes [119]. However, miR-182 plays a key role in pathological osteoclastogenesis and negative regulation of osteoblast differentiation, and its inhibition could provide new strategies for bone protection [57,165].

Finally, **miR-28-3p** expression was inversely correlated with the expression of two BM protective genes (SATB2 and SRCIN1), positively correlated with CXCR4 expression, and correlated with shorter OS.

### 4.3. miRNA-mRNA Associations

We could validate some previously described correlations between miRNAs and mRNA expression of genes involved in the development of BM, although we could not validate other known associations. This can be partly due to the fact that ccRCC is a malignancy with osteolytic BM, whereas most studies have been performed in diseases with osteoblastic BM such as BC and PC. Moreover, only a limited number of samples were included in our study. Nevertheless, our findings on SATB2 in particular are coherent throughout the different parts of this study. This is a DNA binding protein involved in cell differentiation and reprogramming of expression profiles. Its tumor suppressive role has been previously described [72,166], and it holds a critical role in osteoblastogenesis as well [29]. A total of five miRNA associated with shorter TTBM (miR-21-3p, miR-21-5p, miR-28-3p, miR-34c-5p and miR-128-5p) were associated with lower SATB2 levels. A total of two miRNA associated with longer TTBM (miR-204-5p, miR-542-5p) were associated with higher SATB2 expression and could therefore potentially target suppressors of SATB2. SATB2 mRNA levels were correlated with longer TTBM and were also associated with longer OS, since diagnosis and longer PFS and OS since the start of first-line VEGFR-TKI. Several of these associations have previously been described [39,92,94]. Downregulation of SATB2 in ccRCC has previously been associated with metastasis and worse outcome [72].

### 4.4. Limitations and Strengths of Our Study

Our study has several limitations. All patients included eventually developed metastasis, so this is a selection of ccRCC patients. Intratumoral heterogeneity might also be an important issue as miRNA and mRNA extraction was done on different samples of the tumor. As BM were diagnosed through computed tomography scans and not via magnetic resonance imaging, this may result in underestimation. Finally, with miRNA and mRNA expression from tumor tissues, it is not possible to directly demonstrate the impact of miRNA on mRNA expression, only associations.

However, this study has also its strengths. It is an in-depth analysis of the possible impact of miRNAs on the development of BM in ccRCC, whereas most studies have been done in BC and PC. BC and PC are diseases with mainly osteoblastic BM, with different underlying pathophysiology. Not all of our results are specifically for the development of BM. Many of these miRNAs also seem involved in the global process of metastasis, also towards other organs. However, the development of a new therapy directed against BM and debilitating SREs, even if not specifically targeting BM, will still be of benefit for the patient.

## 5. Conclusions

In this study, we present a series of 13 miRNA that might be involved in the pathogenesis of BM in ccRCC and describe similar findings in the literature. These miRNAs could be potential biomarkers or attractive targets for miRNA inhibitors or mimics, which could lead to novel therapeutic possibilities for bone targeted treatment in metastatic malignancies.

## Figures and Tables

**Figure 1 cancers-13-01554-f001:**
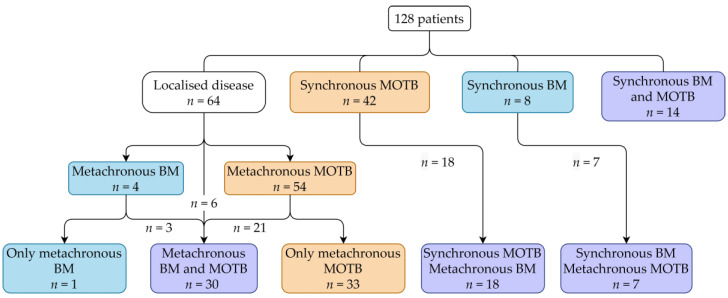
Clinical stage of included patients. Abbreviations: MOTB = metastasis other than bone, BM = bone metastasis.

**Figure 2 cancers-13-01554-f002:**
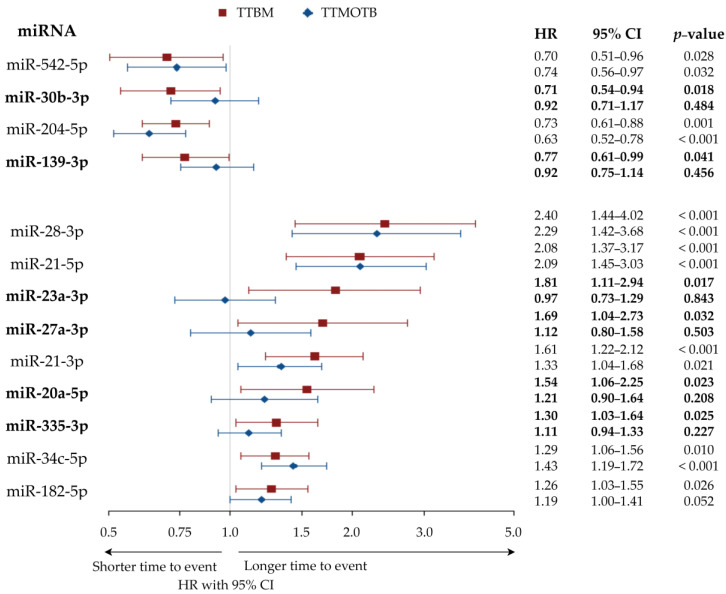
Forest plot displaying HR with 95% CI of TTBM and TTMOTB for 13 miRNA significantly associated with TTBM (univariate Cox proportional hazards model). **Note:** miRNA correlated with TTBM but not with TTMOTB are highlighted in bold. Abbreviations: miRNA = microRNA, TTBM = time to bone metastasis, TTMOTB = time to metastasis other than bone, miR = microRNA, HR = hazard ratio, CI = confidence interval.

**Table 1 cancers-13-01554-t001:** Selected miRNAs and genes.

Selected miRNAs		
let-7a-3p [18,19], let-7a-5p [19,20], let-7g-3p [19], let-7g-5p [19], miR-10b-3p [21,22], miR-10b-5p [21,22], miR-15b-3p [23], miR-15b-5p [23], miR-16-1-3p [24], miR-16-2-3p [24], miR-16-5p [24], miR-17-3p [25], miR-17-5p [26], miR-19a-3p [27], miR-20a-5p [28,29], miR-21-3p [30,31], miR-21-5p [26,30], miR-23a-3p [32,33,34], miR-27a-3p [26,34], miR-28-3p [35], miR-30a-3p [36], miR-30a-5p [36], miR-30b-3p [36], miR-30c-5p [36], miR-30d-3p [36], miR-30d-5p [36], miR-30e-3p [36], miR-30e-5p [36], miR-31-5p [26], miR-33a-5p [37], miR-34a-5p [38], miR-34c-5p [39], miR-125a-3p [26], miR-125a-5p [26], miR-126-5p [26,40], miR-133a-3p [41], miR-135a-5p [42], miR-138-5p [43], miR-139-3p [44,45], miR-139-5p [46], miR-141-3p [47], miR-142-3p [48,49], miR-142-5p [48], miR-143-3p [50,51], miR-143-5p [50], miR-145-3p [50,52], miR-145-5p [50,52], miR-148a-3p [27,53], miR-155-5p [54,55], miR-182-5p [56,57], miR-190a-5p [24], miR-190b [24], miR-200a-3p [58], miR-200a-5p [58], miR-200b-3p [58], miR-200b-5p [58], miR-200c-3p [58], miR-203a-3p [42], miR-204-5p [59], miR-210-3p [55,60], miR-211-5p [59], miR-214-3p [61], miR-214-5p [61], miR-218-5p [62], miR-223-3p [47], miR-223-5p [47], miR-296-5p [27], miR-326 [63], miR-335-3p [18], miR-335-5p [64], miR-378a-3p [24], miR-379-5p [65], miR-409-3p [65], miR-409-5p [65], miR-503-5p [24], miR-542-3p [66], miR-542-5p [67], miR-543 [68]		
**Selected genes**		
**Protective for BM**		
**BMP7**	Bone Morphogenetic Protein 7	Binds BMPR2 receptor on tumor cells and enhances tumor cell dormancy through ERK/MAPK signaling [69].
**BMPR2**	Bone Morphogenetic Protein Receptor Type 2	Promotes tumor cell dormancy in bone [40]. Inversely correlated with prostate BM occurrence [70].
**FOS**	Fos Proto-Oncogene	Transcription factor for osteoclastogenesis [71].
**NOS3**	Nitric Oxide Synthase 3	Decreases invasiveness [68].
**NOTCH1**	Notch Receptor 1	Involved in oncogenesis [40].
**PDCD4**	Programmed Cell Death 4	Represses osteoclastogenesis [71]. Downregulation in NSCLC potentially enhances BM [30].
**SATB2**	Special AT-Rich Sequence-Binding Protein 2	Promotes osteoblastogenesis and bone regeneration [72]. Downregulation is associated with metastasis in ccRCC [72].
**SRCIN1**	SRC Kinase Signaling Inhibitor 1	Inhibits osteoclast differentiation [28].
**SMAD2**	SMAD Family Member 2	Decreases invasiveness. Component of the TGFβ pathway. In BC cells, loss of SMAD2 increases BM potential [73].
**SMAD4**	SMAD Family Member 4	Decreases invasiveness [40,74].
**OPG**	Osteoprotegerin	Decoy receptor preventing RANKL-RANK signaling [9].
**Associated with BM**		
**CD44**	Cluster of differentiator 44	Promotes invasiveness [40,75].
**CDH11**	Cadherin 11	Promotes osteomimicry [36].
**CTGF**	Connective Tissue Growth Factor	Stimulates osteoclasts [40].
**CXCL8**	C-X-C Motif Chemokine Ligand 8	Promotes osteoclastogenesis [76].
**CXCR4**	C-X-C Motif Chemokine Receptor 4	Promotes metastasis to osteogenic niches [40,77].
**DKK1**	Dickkopf WNT Signaling Pathway Inhibitor 1	Osteoclast inhibitor [40].
**ELK1**	ETS Transcription Factor ELK1	Promotes osteoblast differentiation [44]. Inducer of c-FOS proto-oncogene and part of ERK/MSK1/Elk-1/Snail signaling pathway (enhances cancer proliferation) [78].
**IL11**	Interleukin 11	Stimulates osteoclasts [40].
**ITGA3**	Integrin Subunit Alpha 3	Promotes invasiveness [40,75].
**ITGA5**	Integrin Subunit Alpha 5	Involved in invasiveness [36], anchors cancer cells to bone [79].
**ITGB3**	Integrin Subunit Beta 3	Integrin promoting osteomimicry and osteolytic metastases [80].
**RUNX2**	Runt-Related Transcription Factor 2	TF essential in osteoblastogenesis and osteomimicry in PC and BC [42].
**SMAD1**	SMAD Family Member 1	Osteoblast differentiation [81].
**TCF7**	Transcription Factor 7	Involved in BM, part of the Wnt/beta-catenin signaling pathway [38].
**TGIF2**	TGF-Beta-induced transcription factor 2	Pro-osteoclastic factor [82].
**TGFB1**	Transforming Growth Factor Beta 1	Stimulates pro-osteoclastic factor production in cancer cells [40].
**RANKL**	Receptor Activator Of Nuclear Factor Kappa B Ligand	Promotes osteoclastogenesis [10].
**RANK**	Receptor Activator Of Nuclear Factor Kappa B	Osteoclast activation [83].

**Abbreviations**: miRNA = microRNA, miR = microRNA, BM = bone metastasis, NSCLC = non-small cell lung cancer, ccRCC = clear-cell renal cell carcinoma, BC = breast cancer, TF = transcription factor, PC = prostate cancer.

**Table 2 cancers-13-01554-t002:** Patient characteristics.

All Patients	*n* = 128	
Gender: male	86	67%
Median age at diagnosis (years)	62	IQR: 55–69
Median OS after diagnosis (months)	49	IQR: 21–103.25
Median OS after stage IV (months)	34	IQR: 16.5–62.25
**Bone metastasis**		
BM at time of nephrectomy (n)	22	17.2%
Metachronous BM (*n*)	106	82.8%
Median time to metachronous BM (months)	34	IQR: 24.25–100.5
**IMDC risk group at start of first-line therapy**		
Favorable (*n*)	14	11%
Intermediate (*n*)	81	63%
Poor (*n*)	33	26%
**First-line targeted therapy**		
Sunitinib (*n*)	68	53%
Pazopanib (*n*)	31	24%
Sorafenib (*n*)	11	9%
Temsirolimus (*n*)	9	7%
Nivolumab-Ipilimumab (*n*)	6	5%
Other (*n*)	3	2%

Abbreviations: IQR = interquartile range, OS = overall survival (months), BM = bone metastasis, IMDC = International Metastatic RCC Database Consortium.

**Table 3 cancers-13-01554-t003:** Correlations of miRNA with gene expression (Spearman correlation test, *p*-value < 0.05).

miRNAs Associated with Longer TTBM			
**miR-30b-3p**	**BM promoting genes: inverse correlation**		
	RANKL	rho = −0.33; *p* = 0.002	miR-30 family inhibits BM in BC by targeting **CXCL8** [36]
	ITGA3	rho = −0.31; *p* = 0.003	
	TCF7	rho = −0.25; *p* = 0.018	
	CD44	rho = −0.24; *p* = 0.021	
	**CXCL8**	**rho = −0.21; *p* = 0.045**	
**miR-139-3p**	**BM protective genes: positive correlation**		
	NOTCH1	rho = 0.29; *p* = 0.005	
	BMPR2	rho = 0.27; *p* = 0.011	
	**BM promoting genes: inverse correlation**		
	TCF7	rho = −0.23; *p* = 0.027	
	ITGA3	rho = −0.22; *p* = 0.038	
miR-204-5p	**BM protective genes: positive correlation**		
	FOS	rho = 0.28; *p* = 0.007	
	BMPR2	rho = 0.32; *p* = 0.003	
	SATB2	rho = 0.34; *p* = 0.001	
	SMAD4	rho = 0.35; *p* = 0.001	
	**BM promoting genes: inverse correlation**		
	CD44	rho = −0.55; *p* < 0.001	miR-204 regulates **RUNX2** expression and MSC differentiation [85].
	RANKL	rho = −0.42; *p* < 0.001	Sponging of miR-204-5p by lncRNA SNHG4 upregulates **RUNX2** and promotes tumor progression in RCC [86].
	ITGA3	rho = −0.42; *p* < 0.001	miR-204-5p inhibits **TGF-β**-induced IL11 production in BM cells of BC [59].
	**RUNX2 °**	**rho = −0.41; *p* < 0.001**	miR-204 inhibits growth and motility of CRC cells by **CXCL8** downregulation [87].
	ITGA5	rho = −0.36; *p* = 0.001	
	CDH11 °	rho = −0.39; *p* < 0.001	
	**TGFB1**	**rho = −0.33; *p* = 0.002**	
	**CXCL8**	**rho = −0.3; *p* = 0.005**	
	TCF7	rho = −0.36; *p* < 0.001	
	CTGF	rho = −0.26; *p* = 0.014	
miR-542-5p	**BM protective genes: positive correlation**		
	SATB2	rho = 0.34; *p* = 0.001	miR-542-5p increases **SMAD2** expression levels in ICU weakness [88].
	**SMAD2**	**rho = 0.29; *p* = 0.006**	
	**BM promoting genes: inverse correlation**		
	TCF7	rho = −0.37; *p* < 0.001	
	RUNX2	rho = −0.31; *p* = 0.003	
	CD44	rho = −0.32; *p* = 0.002	
	CXCR4	rho = −0.21; *p* = 0.044	
**miRNAs Associated With Shorter TTBM**			
**miR-21-5p**	**BM protective genes: inverse correlation**		
	SATB2	rho = −0.22; *p* = 0.04	
	SMAD4	rho = −0.24; *p* = 0.026	
	**BM promoting genes: positive correlation**		
	TCF7	rho = 0.27; *p* = 0.011	
	CD44 *	rho = 0.25; *p* = 0.018	
	ITGA3	rho = 0.29; *p* = 0.005	
miR-21-3p	**BM protective genes: inverse correlation**		
	**FOS**	**rho = −0.45; *p* < 0.001**	miR-21 deficiency results in **cFOS** upregulation in periodontal tissues [89].
	**BMPR2**	**rho = −0.42; *p* < 0.001**	**BMPR2** is directly targeted by miR-21 in PC cells [90].
	SATB2	rho = −0.31; *p* = 0.003	miR-21-3p is involved in proliferation and invasion through **SMAD4**/Erk signaling in CRC [31].
	**SMAD4**	**rho = 0.22; *p* = 0.043**	
	**BM promoting genes: positive correlation**		
	ITGA3	rho = 0.42; *p* < 0.001	**TGFB1** signaling increases miR-21 expression in renal fibrosis [91].
	RANKL	rho = 0.38; *p* < 0.001	
	CD44 *	rho = 0.37; *p* < 0.001	
	**TGFB1**	**rho = 0.25; *p* = 0.018**	
	CXCL8 *	rho = 0.22; *p* = 0.043	
miR-28-3p	**BM protective genes: inverse correlation**		
	SATB2	rho = −0.24; *p* = 0.022	
	SRCIN1	rho = −0.22; *p* = 0.038	
	**BM promoting genes: positive correlation**		
	CXCR4	rho = 0.24; *p* = 0.025	
miR-34c-5p	**BM protective genes: inverse correlation**		
	**SATB2 °**	**rho = −0.23; *p* = 0.033**	miR-34s inhibit osteoblast proliferation and differentiation in mouse by targeting **SATB2** [92].
	FOS	rho = −0.26; *p* = 0.015	**SATB2** targeted by miR-34c-5p suppresses proliferation and metastasis attenuating EMT in CRC [39].
	**BM promoting genes: positive correlation**		
	CD44	rho = 0.3; *p* = 0.004	
	ITGA3	rho = 0.38; *p* < 0.001	
	RANK	rho = 0.21; *p* = 0.047	
	TGFB1	rho = 0.26; *p* = 0.015	
**miR-23a-3p**	**BM protective genes: positive correlation**		
	BMPR2 *	rho = 0.23; *p* = 0.027	
	**BM promoting genes: inverse correlation**		
	CXCL8 *	rho = −0.25; *p* = 0.017	
	ELK1	rho = −0.24; *p* = 0.021	
**miR-20a-5p**	**BM protective genes: inverse correlation**		
	OPG *	rho = −0.21; *p* = 0.044	
	**BM protective genes: positive correlation**		
	PDCD4 °	rho = 0.28; *p* = 0.009	
	**BM promoting genes: inverse correlation**		
	ITGB3	rho = −0.23; *p* = 0.03	
	**BM promoting genes: positive correlation**		
	IL11	rho = 0.27; *p* = 0.011	**RUNX2** is positively correlated with miR-20a-5p in adipose SC during osteogenic differentiation [93].
	TCF7	rho = 0.26; *p* = 0.012	
	**RUNX2**	**rho = 0.22; *p* = 0.035**	
**miR-335-3p**	**BM protective genes: inverse correlation**		
	OPG	rho = −0.23; *p* = 0.03	
	**BM promoting genes: positive correlation**		
	CD44	rho = 0.31; *p* = 0.003	
	ITGA5	rho = 0.29; *p* = 0.007	
	RANKL	rho = 0.22; *p* = 0.04	
	CDH11	rho = 0.36; *p* = 0.001	
	ITGA3	rho = 0.27; *p* = 0.011	
	TGFB1	rho = 0.29; *p* = 0.006	
	TCF7	rho = 0.35; *p* = 0.001	
	RUNX2	rho = 0.3; *p* = 0.004	
**miR-182-5p**	**BM protective genes: inverse correlation**		
	**SATB2 *^,^°**	**rho = −0.29; *p* = 0.006**	microRNA-182 targets **SATB2** to promote CRC proliferation and metastasis [94].
	BMPR2 *	rho = −0.28; *p* = 0.009	
	OPG	rho = −0.28; *p* = 0.007	
	**BM promoting genes: positive correlation**		
	CD44	rho = 0.37; *p* < 0.001	
	CDH11 *	rho = 0.25; *p* = 0.02	
	ITGA3	rho = 0.37; *p* < 0.001	
	IL11 *	rho = 0.21; *p* = 0.046	
	CXCL8	rho = 0.32; *p* = 0.002	
	TCF7	rho = 0.34; *p* = 0.001	
	RUNX2	rho = 0.27; *p* = 0.011	
***miR-27a-3p***	**BM protective genes: inverse correlation**		
	BMP7 *	rho = −0.29; *p* = 0.007	
	**BM protective genes: positive correlation**		
	PDCD4	rho = 0.23; *p* = 0.031	
	**BM promoting genes: inverse correlation**		
	CXCL8 *	rho = −0.26; *p* = 0.013	

**BM protective genes**: SATB2, BMP7, SMAD4, PDCD4, BMPR2, OPG, NOS3, SMAD2, NOTCH1, FOS, SRCIN1. **BM promoting genes**: TCF7, DKK1, CD44, ITGA3, CTGF, IL11, CXCL8, RANKL, RUNX2, CDH11, TGFB1, CXCR4, SMAD1, TGIF2, RANK, ELK1, ITGB3, ITGA5. *** = listed in DIANA-TarBase v.8 of validated miRNA-target pairs** [16]. **° = predicted miRNA target through miRDB.org** [17]. **Note:** miRNA correlated to TTBM and not to TTMOTB are highlighted in bold. Correlations with similar literature findings are highlighted in bold. **Abbreviations**: miRNA = microRNA, TTBM = time to bone metastasis, miR = microRNA, BM = bone metastasis, BC = breast cancer, MSC = mesenchymal stem cells, lncRNA = long non-coding RNA, SNHG4 = small nucleolar RNA host gene 4, RCC = renal cell carcinoma, CRC = colorectal cancer, PC = prostate cancer, EMT = epithelial-mesenchymal transition, RCC = renal cell carcinoma, SC = stem cells.

**Table 4 cancers-13-01554-t004:** Correlations of 13 miRNA involved in bone metastasis with OS since diagnosis and PFS and OS on 1st line VEGFR-TKI therapy (univariate Cox regression model).

**miRNA**	**OS Since Diagnosis**		**PFS on VEGFR-TKIs**		**OS on VEGFR-TKIs**	
	HR (95% CI)	*p*-value	HR (95% CI)	*p*-value	HR (95% CI)	*p*-value
**Longer time to bone metastasis**						
**miR-204-5p**	**0.73 (0.65–0.82)**	**<0.001**	**0.84 (0.74–0.94)**	**0.003**	**0.81 (0.72–0.92)**	**0.001**
miR-30b-3p	0.93 (0.78–1.10)	0.386	1.07 (0.88–1.31)	0.496	0.99 (0.81–1.21)	0.921
miR-542-5p	0.85 (0.69–1.05)	0.133	0.96 (0.77–1.20)	0.728	0.89 (0.71–1.11)	0.296
miR-139-3p	0.86 (0.73–1.00)	0.057	0.91 (0.76–1.09)	0.321	0.87 (0.73–1.03)	0.105
**Shorter time to bone metastasis**						
**miR-21-5p**	**1.69 (1.31–2.18)**	**<0.001**	1.14 (0.88–1.47)	0.327	**1.29 (1.01–1.65)**	**0.043**
**miR-21-3p**	**1.50 (1.31–2.18)**	**<0.001**	**1.24 (1.04–1.48)**	**0.016**	**1.35 (1.13–1.61)**	**0.001**
miR-28-3p	1.38 (0.98–1.93)	0.064	1.01 (0.70–1.47)	0.959	1.04 (0.72–1.50)	0.823
**miR-34c-5p**	**1.31 (1.16–1.48)**	**<0.001**	**1.19 (1.05–1.35)**	**0.007**	**1.27 (1.11–1.44)**	**<0.001**
miR-23a-3p	1.05 (0.83–1.33)	0.697	0.98 (0.75–1.30)	0.903	1.01 (0.76–1.34)	0.953
miR-20a-5p	1.12 (0.91–1.39)	0.279	0.95 (0.77–1.19)	0.680	0.94 (0.75–1.18)	0.598
**miR-335-3p**	**1.22 (1.06–1.41)**	**0.005**	1.10 (0.95–1.26)	0.193	1.11 (0.96–1.28)	0.155
**miR-182-5p**	**1.25 (1.09–1.43)**	**0.002**	1.11 (0.96–1.29)	0.145	**1.16 (1.01–1.34)**	**0.040**
miR-27a-3p	1.13 (0.87–1.47)	0.351	0.87 (0.65–1.17)	0.365	1.03 (0.77–1.39)	0.831

Abbreviations: miRNA = microRNA; BM = bone metastasis; PFS = progression free survival; OS = overall survival; VEGFR-TKI = vascular endothelial growth factor receptor tyrosine kinase inhibitor; HR = hazard ratio; CI = confidence interval; miR = microRNA. Significant correlations (*p*-value < 0.05) are highlighted in bold.

## Data Availability

The data presented in this study are available on request from the corresponding author. The data are not publicly available due to privacy concerns.

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
