# Peer review of "MicroRNAs Possibly Involved in the Development of Bone Metastasis in Clear-Cell Renal Cell Carcinoma"

_cancers, 2021, doi:10.3390/cancers13071554_

Round 1

Reviewer 1 Report

The manuscript entitled “MicroRNAs possibly involved in the development of bone metastasis in clear-cell renal cell carcinoma” is an interesting literature revision on the role of skeletal metastasis in renal carcinoma.

The study is well written and balance, so far it should be published after some minor revisions: 

  • In the text there few mistyping. Please revise and correct them
  • Please revise all abbreviation in the main text
  • In the abstract, it is unclear the type of the study. Please specify this topic.
  • Reference 1 should be updated
  • Please specify the name of Ethics Committee and the number of its approval.
  • Correlation of miRNA with clinical outcome (Table 4 and Table S4-S5-S6). Table referring should report in the text
  • DISCUSSION – Please stress the incidence and patients characteristics of renal cancer patients with bone metastasis (Authors can refer to: Santini et al Natural history of malignant bone disease in renal cancer: final results of an Italian bone metastasis survey. PLoS One. 2013 Dec 30;8(12):e83026; Santoni et al, Bone metastases in patients with metastatic renal cell carcinoma: are they always associated with poor prognosis? J Exp Clin Cancer Res. 2015 Feb 5;34(1):10)

Author Response

Thank you for your thorough and constructive comments. We made the following adjustments:

Point 1: In the text there few mistyping. Please revise and correct them

Response 1: I have corrected the spelling mistake (contect instead of content) on line 112. 

Point 2: Please revise all abbreviation in the main text

Response 2: I have corrected the abbreviations on line 171 (IQR), 189 (HR and CI), 221 to 226, 332 (SCC) and 362 (CRC).

Point 3: In the abstract, it is unclear the type of the study. Please specify this topic.

Response 3: I have added that this is a retrospective analysis in the abstract. 

Point 4: Reference 1 should be updated

Response 4:  I have added the reference "Santini et al Natural history of malignant bone disease in renal cancer: final results of an Italian bone metastasis survey. PLoS One. 2013 Dec 30;8(12):e83026" to the first sentence.

Point 5: Please specify the name of Ethics Committee and the number of its approval.

Response 5: I have specified the name of the EC and the approval number in the materials and methods section.   

Point 6: Correlation of miRNA with clinical outcome (Table 4 and Table S4-S5-S6). Table referring should report in the text

Response 6:  I added a referral to Table S5 and S6 in the text. 

Point 7: DISCUSSION – Please stress the incidence and patients characteristics of renal cancer patients with bone metastasis (Authors can refer to: Santini et al Natural history of malignant bone disease in renal cancer: final results of an Italian bone metastasis survey. PLoS One. 2013 Dec 30;8(12):e83026; Santoni et al, Bone metastases in patients with metastatic renal cell carcinoma: are they always associated with poor prognosis? J Exp Clin Cancer Res. 2015 Feb 5;34(1):10)

Response 7:  I have added further context on the incidence and patients characteristics at the start of the discussion (line 258-262), whilst referring to the references you kindly provided. 

We hope these revisions meet your expectations. 

Reviewer 2 Report

Kinget L et al present an interesting work regarding a complex and still quite unexplored topic. It is redacted and explained in a clear and logical fashion, although the great amount of information exposed difficults the comprehension at some points.

I have some concerns regarding the choice of "time to bone metastases" as the main variable, as well as the exclusion for the analysis of those patients who already presented bone metastases at the time of diagnosis, as this probably leads to the selection of patients with a better prognosis. Other variables as the presence or not of bone metastases during the course of the disease present other limitations, altough it would be interesting to know if any association also was found.

The main limitation of the study is that it presents associations and does not prove any mechanistics. However, as a exploratory work I think it merits recognition and it is worthy of publication.

Author Response

Thank you for your thorough review and constructive comments. 

Point 1:

I have some concerns regarding the choice of "time to bone metastases" as the main variable, as well as the exclusion for the analysis of those patients who already presented bone metastases at the time of diagnosis, as this probably leads to the selection of patients with a better prognosis. Other variables as the presence or not of bone metastases during the course of the disease present other limitations, although it would be interesting to know if any association also was found.

Response 1:  

As the main variables we studied are time-related (time to bone metastasis, time to metastasis other than bone), we have chosen to exclude the patients with synchronous bone metastasis as these time events were not clear for those patients. A slight selection bias for patients with better prognosis is inevitable. 

Point 2: 

The main limitation of the study is that it presents associations and does not prove any mechanistics. However, as a exploratory work I think it merits recognition and it is worthy of publication.

Response 2: 

This is indeed a limitation and further research into these miRNA using cell lines is definitely required. To highlight this, we have underlined the exploratory nature of this study and elaborated on this in the limitation section. These findings do add to the existing literature on miRNA's in bone metastasis in ccRCC patients, so we hope it can guide future research into the underlying mechanisms of these miRNA's in bone metastasis.

Hopefully these clarifications are satisfactory.

Reviewer 3 Report

This manuscript provided very interesting findings on clear-cell renal cell carcinoma (ccRCC). Experiments were conducted to determine the miRNA and mRNA expression in ccRCC tumors and utilized the clinical data on the patients to determine whether specific miRNA are associated with metastasis in ccRCC. Authors demonstrated six microRNA as potential biomarkers for bone metastasis of ccRCC. Among these six miRNA four implicate in higher risk (miR-23a-3p, miR-27a-3p, miR-20a-5p and miR-355-3p) and two in protective (miR-30b-3p and miR-139-3p).  Experimental design is well thought to evaluate miRNA as potential biomarkers and therapeutic targets in ccRCC and manuscript is well written. I have some minor comments.

  1. Section 2.5 line 127, provide subset patients number, same in line 129 provide subset of patients number.
  2. In Figure 2, for TTMOTB light blue line is not clearly seen like dark red line for TTBM. Either change the color of TTMOTB or make it darker/think so that it will be clearly seen.
  3. Discussion looks very lengthy, but it is good that comprehensively discussed all the findings of this work. I would recommend to add one or two sentences on the 6 miRNA associated with metastasis. Many of these miRNA have been reported as circulating microRNA in biofluids (serum/plasma). These circulating microRNA which served as potential biomarkers can be evaluated non-invasively in sera of ccRCC patients.

Author Response

Thank you for your thorough and constructive comments. 

We have made the following changes:

Point 1: Section 2.5 line 127, provide subset patients number, same in line 129 provide subset of patients number.

Response 1: I added the respective patients numbers (106 and 72) on those lines. 

Point 2: In Figure 2, for TTMOTB light blue line is not clearly seen like dark red line for TTBM. Either change the color of TTMOTB or make it darker/think so that it will be clearly seen.

Response 2: I changed the color of TTMOTB to a darker blue, so it is more clearly seen.

Point 3: Discussion looks very lengthy, but it is good that comprehensively discussed all the findings of this work. I would recommend to add one or two sentences on the 6 miRNA associated with metastasis. Many of these miRNA have been reported as circulating microRNA in biofluids (serum/plasma). These circulating microRNA which served as potential biomarkers can be evaluated non-invasively in sera of ccRCC patients.

Response 3: For miR-139-3p, miR-23a, miR-27a-3p and miR-20a-5p, the available evidence was most substantial and this was added to the manuscript (line 314-316, line 345-347, line 359-361 and line 376-378).

Hopefully you find these revisions satisfactory.